# A novel hole performance index to evaluate the hole geometry and drilling time in the electrochemical drilling process

N. Tosun[1], H. B. Özerkan[2]*, C. Çoğun[3], Robert Cep[4]

1 Mechanical Engineering Department, Firat University, Elaziğ, Turkey, 2 Mechanical Engineering Department, Gazi University, Ankara, Turkey, 3 Mechatronics Engineering Department, Çankaya University, Ankara, Turkey, 4 Faculty of Mechanical Engineering, VSB—Technical University of Ostrava, Ostrava, Czechia Republic

* ozerkan@gazi.edu.tr

## Abstract

Electrochemical Drilling (ECD) is an unconventional method aimed at creating holes in metallic workpieces characterized by high hardness and complex structures. This study analyzes the influence of process variables, including machining voltage, electrolyte concentration, electrode rotational speed, electrolyte flushing pressure, and workpiece material, on the novel hole performance index (HPI) in electrical discharge machining (ECD). The HPI was identified as a suitable metric for simultaneously evaluating hole geometry and drilling time across various machining parameters and workpiece materials. The analysis of variance (ANOVA) method was employed to determine the significance of each machining parameter and workpiece material on the HPI. The research employed signal-to-noise ratio analysis to identify the optimal machining parameters. The findings demonstrated that the workpiece material and machining voltage were significant factors influencing HPI. The validation tests demonstrated that the proposed statistical method can significantly reduce HPI.

## Introduction

Electrochemical hole drilling (ECD) is a nonconventional manufacturing approach developed to create holes with various cross-sections. The ECD method involves separating the cathode electrode from the anode workpiece by a small gap. An electrical current is applied to the electrode to dissolve the anode through anodization. An electrolyte is flushed from the center of the insulated electrode towards the workpiece to remove debris from the processing medium and cool the medium during operation (Fig 1).

Gaps identified in literature regarding performance outputs reflecting hole geometry and drilling time in the ECD process are presented here. The average diameter [1–10] and radial overcut [10–22] were the most commonly used performance outputs in ECD research. Other outputs include inlet/exit diameters [5, 10, 11, 15, 23–25], taper (conicity) [4, 10, 12, 13, 18, 26], material removal rate [14, 16–19, 21–23, 27], roundness [7–9], wall surface roughness [8, 9, 19, 26], depth [6, 24, 28, 29], drilling stability [3, 11, 23, 30], drilling time [18, 23, 26],

**Funding:** This article was co-funded by the European Union under the REFRESH—Research Excellence For Region Sustainability and High-tech Industries project number CZ.10.03.01/00/22_003/0000048 via the Operational Programme Just Transition and has been done in connection with project Students Grant Competition SP2024/087 "Specific Research of Sustainable Manufacturing Technologies" financed by the Ministry of Education, Youth and Sports and Faculty of Mechanical Engineering VSB-TUO. The article has been done in connection with the project Students Grant Competition SP2024/087 "Specific Research of Sustainable Manufacturing Technologies," financed by the Ministry of Education, Youth and Sports and Faculty of Mechanical Engineering VSB-TUO.

**Competing interests:** The authors have declared that no competing interests exist.

**Abbreviations:** $d_e$, Electrode diameter (mm; h, Hole depth (mm; $n_e$, Electrode rotational speed (rpm; $t_m$, Hole drilling time (min; $C_e$, Electrolyte salt concentration (g/; $A_e$, Electrode cross-sectional area (mm$^2$); $A_{hole}$, Hole cross-sectional area (mm$^2$; $L_r$, Length of the right-side wall profile (mm; $L_l$, Length of the left-side wall profile (mm; $P_f$, Electrolyte flushing pressure (bar; $V_m$, Machining voltage (V); ECD, Electrochemical drilling; HACI, Hole shape conformity index; HGI, Hole geometry index; HPI, Hole performance index; HWPCI, Hole wall profile conformity index; PM, Powder metallurgy.

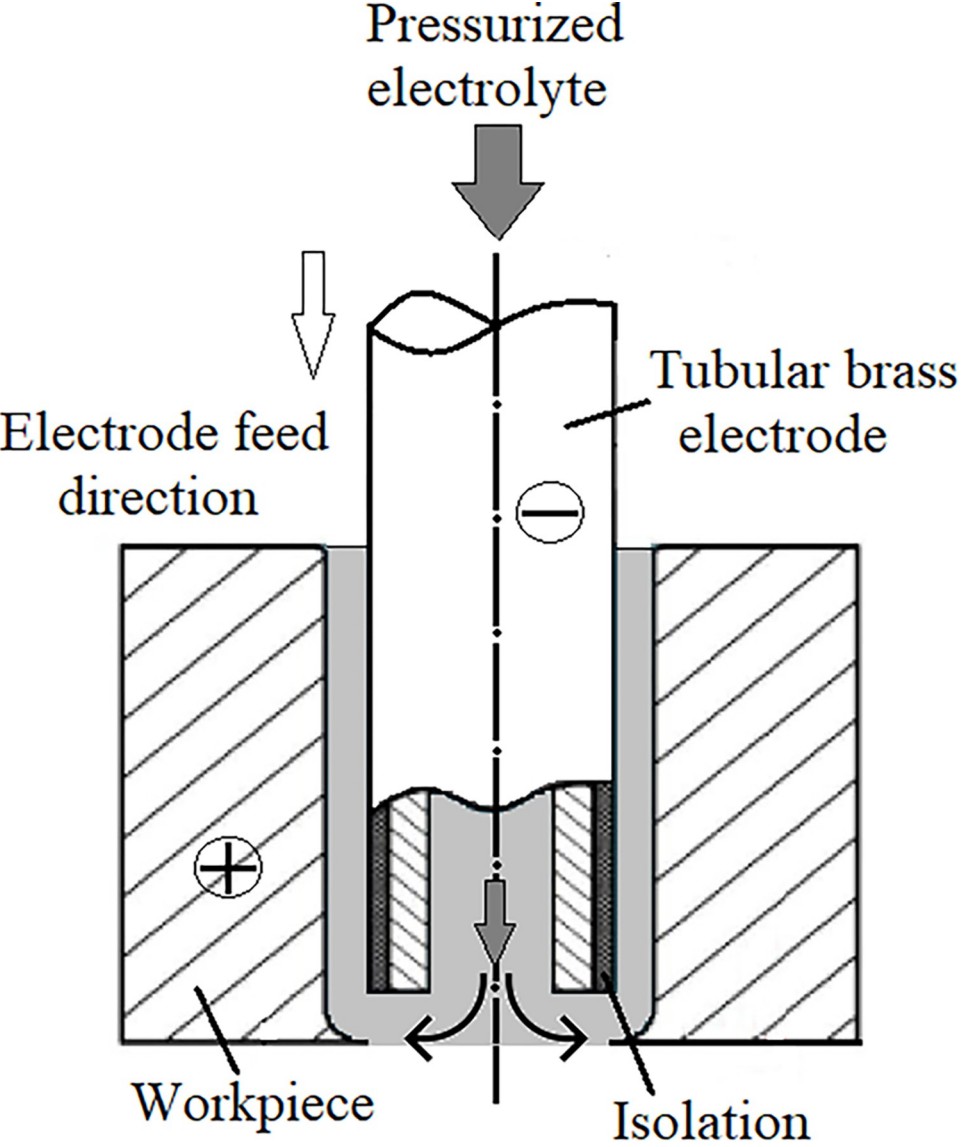

**Fig 1. Schematic illustration of the ECD process utilizing a tubular electrode.**

machining gap [26, 28, 31], machining current [3, 30], electrode wear [26, 31], wall area [29], wall profile length [29], bottom geometry (for blind holes) [31], and wall delamination (for composite workpieces) [14] of the hole. The performance outputs related to hole dimensions and geometry were examined using tools such as optical microscopes, laser scanning microscopes, digital microscopes, 3D-coordinate measuring machines, and image processing software. These outputs were sometimes combined to create new indexes defining hole quality and drilling performance in ECD. The study by Bilgi et al. [16, 17] introduced a hole quality factor (HQF), the multiplication of depth-averaged radial overcut and the cutting rate to electrode feed rate ratio. The Ayhan et al. study [29] used indexes to evaluate the geometrical characteristics of the hole, including the ratio of the depth of the hole to the set depth (HDC), the ratio of the cross-sectional area of the hole to the electrode (HCAC) and the ratio of cross-sectional profile length of the hole to the electrode (HPLC). The HCAC index was first introduced

by one of the authors of this study [32] to represent the geometric quality of the machined cavity in die-sinking electric discharge machining. However, no study has been conducted on defining novel performance indexes that combine hole geometry with drilling time. Furthermore, statistical techniques have not been used to assess the impact of process parameters and drilling time on the new performance indexes.

During deep electrochemical drilling processes, the accumulation of ionic and solid residues from electrolysis causes instabilities in the processing medium, leading to temporary variations in the machining current and workpiece removal rate [35]. This can result in the formation of irregularly shaped holes with non-uniform and wavy walls. This type of wall, caused by electrochemical machining instabilities, has not been considered when assessing the geometric quality of the holes.

The drilled workpiece materials studied in the research are challenging to machine conventionally. These materials include Ni [13, 23], Ni alloys [2, 16, 17], stainless steels (typically SS 304) [1, 3, 5, 12, 15, 20–22, 24, 25, 27], Inconel [4, 33], Fe-Ni alloy [18], Ti alloys (typically Ti6Al4V) [8, 9, 28], Cr-Ni alloy [11], HSS [29], $ZrO_2$ ceramic [7], WC-Co alloy [26], hard steel alloys [30], valve steels [19] and $Al-TiB_2$ composite [14]. As exemplified by Hadfield Steel, there is still considerable scope for using materials such as hard-to-machine parts due to strain hardening under cutting forces. Furthermore, powder metallurgy (PM) parts, which exhibit distinctive anodization characteristics due to their unique compacted and sintered powder structure, are worth investigating.

### Objectives

This study aims to investigate the performance of a novel Hole Performance Index (*HPI*) in evaluating hole geometry and drilling time concurrently in ECD. Machining settings will be optimized for minimum *HPI* across different workpiece materials and machining parameters, including machining voltage, electrode rotational speed, electrode flushing pressure, and electrolyte concentration. The workpiece materials to be evaluated are AISI 1040 steel, PM part, and Hadfield steel. The impact of machining parameters on *HPI* will be evaluated using Analysis of Variance (ANOVA) to interpret the experimental findings statistically. Signal-to-noise (S/N) ratio analysis will be utilized to obtain the parameters for the minimum *HPI*.

## Materials and methods

### Materials

The experiments were conducted using the setup developed in a previous study (Fig 2) by the authors [34]. The setup head features vertical axis movement with a square screw transmission mechanism. The electrode is fed towards the workpiece by a servo motor, controlled by the current in the machining medium. The electrode rotation is powered by a Maxon A/max15 servo motor, managed by the MIP 10 driver controller. The PC program written in DELPHI controls the electrode feed motion. The electrode is attached to the electrolyte pressure head using a mandrel and sealing gasket. A ceramic guide near the electrode tip prevents lateral deviation during rotation. The machining gap is monitored by a current sensor and adjusted using current values. An XRF 1200 DC power supply provides the specified machining current.

A NaCl-water solution was used as the electrolyte due to its high electrochemical reactivity, affordability and availability. 0.5 mm diameter, 300 mm length brass (70% Cu, 30% Zn, by weight) hollow electrodes with 0.18 mm diameter center hole were used to drill through holes in workpiece materials. The outer cylindrical surface of the electrode is coated with cyanoacrylate to prevent taper formation caused by excessive machining of the hole wall as the electrode

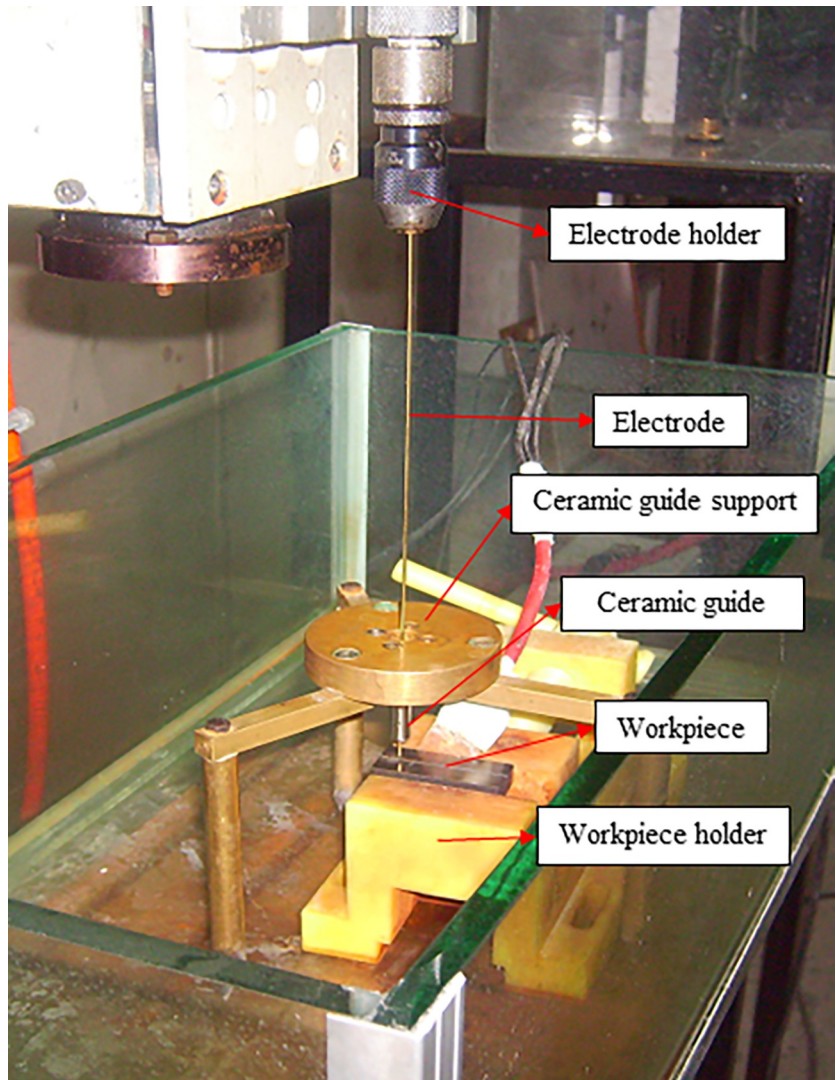

**Fig 2. Electrochemical drilling setup [34].**

penetrates. The electrolyte was flushed through the electrode hole. A piston pump was used to pressurize and flush the electrolytic.

As workpiece materials, AISI 1040 steel specimens of 50 mm x 10 mm x 8 mm, Hadfield steel specimens of 50 mm x 10 mm x 8 mm and sinter-hardened FLN2-4405 PM steel specimens of 50 mm x 7 mm x 14 mm were used in the experiments. The composition of AISI 1040 steel was 0.75% Mn, 0.38% C, 0.20% Si, 0.07% Cr, 0.01% Mo, 0.1% Ni, 0.15% Cu by weight, balance Fe. The composition of Hadfield steel was 13.6% Mn, 1.08% C, 0.72% Cr, 0.62% Si, 0.3% Ni, 0.26% Mo, and 0.18% Cu by weight, balance Fe. The sinter-hardened FLN2-4405 PM steel had a composition of 1% Ni, 0.65% Mo, 0.4% C and 0.05% Mn by weight, balance Fe.

## Methods

**Experimental procedure.**   In the experiments, a through hole was drilled at the junction of two workpieces, which allowed for the study of the hole geometries without cutting the workpieces in half. The experiments tested four machining parameters: machining voltage $V_m$,

electrolyte concentration $C_e$, electrode rotational speed $n_e$ and electrolyte flushing pressure $P_f$. The machining settings used are shown in Table 1. Prior to conducting the experiments, preliminary tests were carried out to determine the machining parameters. It was observed that when the machining voltage settings exceeded 6.5 V, the electrode front surface would adhere to the workpiece due to frequent short-circuit formations at the front gap. Conversely, at voltage settings below 4.5 V, the drilling process took considerably longer. As a result, three different machining voltage settings within the determined range were chosen, as voltage is a crucial parameter in ECD. When the electrode rotational speed fell below 200 rpm, electrolyte flushing pressure dropped below 20 bar, or electrolyte concentrations exceeded 150 g/l, the reaction products would clog the radial gap, resulting in irregular hole walls. On the other hand, with electrode rotational speeds exceeding 400 rpm and electrolyte pressure surpassing 40 bar, the machining became unstable due to the turbulent behavior of the electrolyte in the gap. Additionally, electrolyte concentrations below 100 g/l led to excessively long drilling times. Thus, three electrolyte concentration settings within the defined range were selected, as the drilling process is susceptible to the electrolyte concentration. The electrode rotational speed and electrolyte flushing pressure settings were limited to two levels as they have a lesser impact on the material removal rate.

A full factorial design of experiments was used to evaluate the effect of workpiece materials and machining parameters on *HPI* (Table 2). The drilling experiment was performed three times to ensure the reliability of the results and the measurements were averaged.

**Definition of hole geometry and drilling performance evaluation indexes.** One of the distinctive aspects of this study is the development of novel indexes, listed below, for evaluating hole geometry and drilling performance.

The Hole Shape Conformity Index *HACI* is a measure of the conformity between the hole cross-sectional area $A_{hole}$ and the electrode cross-sectional area $A_e$. $A_e$ is the product of the electrode diameter $d_e$ and the hole depth $h$ (Fig 3). The *HACI* formula is

$$HACI = \frac{A_{hole}}{A_e} \qquad (1)$$

In the ECD process, the *HACI* is always greater than one because the $A_{hole}$ is always larger than the $A_e$ due to the electrochemical dissolution of the sidewall of the hole. The *HACI* approaches 1 when the $A_{hole}$ of the hole is close to $d_e$.

The Hole Wall Profile Conformity Index *HWPCI* measures the degree of waviness of the hole walls, indicating the deviation of the hole wall geometry from a perfect cylinder geometry. The length of the right- and left-side wall profiles of the drilled hole cross-section are represented by $L_r$ and $L_l$, respectively (Fig 3). The formula for *HWPCI* is

$$HWPCI = \frac{L_l + L_r}{2h} \qquad (2)$$

**Table 1. Experimental settings.**

| Parameters | Level | | |
|---|---|---|---|
| | **1** | **2** | **3** |
| Workpiece material $M$ | AISI 1040 | Hadfield | PM |
| Machining voltage $V_m$ (V) | 4.5 | 5.5 | 6.5 |
| Electrode rotational speed $n_e$ (rpm) | 200 | 400 | - |
| Electrolyte flushing pressure $P_f$ (bar) | 20 | 40 | - |
| Electrolyte concentration $C_e$ (g/L) | 100 | 125 | 150 |

**Table 2. Experimental design and *HPI* results.**

| Machining voltage $V_m$ (V) | Electrode rotational speed $n_e$ (rpm) | Electrolyte flushing pressure $P_f$ (bar) | Electrolyte concentration $C_e$ (g/L) | HPI | | |
|---|---|---|---|---|---|---|
| | | | | AISI 1040 steel | Hadfield steel | PM steel |
| 4.5 | 200 | 20 | 100 | 7.85 | 21.68 | 11.61 |
| 4.5 | 400 | 20 | 100 | 7.73 | 23.34 | 9.83 |
| 4.5 | 200 | 40 | 100 | 7.72 | 20.59 | 9.20 |
| 4.5 | 400 | 40 | 100 | 7.55 | 20.10 | 7.51 |
| 4.5 | 200 | 20 | 125 | 8.71 | 22.82 | 6.81 |
| 4.5 | 400 | 20 | 125 | 6.88 | 14.85 | 5.98 |
| 4.5 | 200 | 40 | 125 | 7.12 | 17.50 | 5.81 |
| 4.5 | 400 | 40 | 125 | 7.14 | 16.81 | 5.80 |
| 4.5 | 200 | 20 | 150 | 7.25 | 21.03 | 6.68 |
| 4.5 | 400 | 20 | 150 | 6.24 | 20.41 | 5.45 |
| 4.5 | 200 | 40 | 150 | 7.16 | 16.11 | 5.97 |
| 4.5 | 400 | 40 | 150 | 6.09 | 21.51 | 5.94 |
| 5.5 | 200 | 20 | 100 | 6.30 | 13.62 | 5.14 |
| 5.5 | 400 | 20 | 100 | 5.96 | 20.47 | 5.29 |
| 5.5 | 200 | 40 | 100 | 7.16 | 10.75 | 5.18 |
| 5.5 | 400 | 40 | 100 | 7.60 | 10.89 | 4.07 |
| 5.5 | 200 | 20 | 125 | 5.92 | 10.81 | 4.11 |
| 5.5 | 400 | 20 | 125 | 4.94 | 8.61 | 4.82 |
| 5.5 | 200 | 40 | 125 | 6.46 | 9.64 | 5.04 |
| 5.5 | 400 | 40 | 125 | 6.24 | 9.94 | 3.55 |
| 5.5 | 200 | 20 | 150 | 6.16 | 8.26 | 3.82 |
| 5.5 | 400 | 20 | 150 | 4.95 | 9.02 | 2.95 |
| 5.5 | 200 | 40 | 150 | 5.29 | 9.40 | 3.75 |
| 5.5 | 400 | 40 | 150 | 5.22 | 8.81 | 3.23 |
| 6.5 | 200 | 20 | 100 | 5.60 | 7.56 | 4.60 |
| 6.5 | 400 | 20 | 100 | 6.76 | 7.37 | 3.48 |
| 6.5 | 200 | 40 | 100 | 5.52 | 8.42 | 4.79 |
| 6.5 | 400 | 40 | 100 | 4.78 | 8.17 | 4.87 |
| 6.5 | 200 | 20 | 125 | 5.64 | 8.56 | 4.99 |
| 6.5 | 400 | 20 | 125 | 5.82 | 11.81 | 3.72 |
| 6.5 | 200 | 40 | 125 | 5.13 | 8.51 | 3.94 |
| 6.5 | 400 | 40 | 125 | 5.81 | 8.81 | 3.12 |
| 6.5 | 200 | 20 | 150 | 4.69 | 9.22 | 4.54 |
| 6.5 | 400 | 20 | 150 | 4.51 | 8.10 | 3.77 |
| 6.5 | 200 | 40 | 150 | 4.40 | 9.42 | 3.91 |
| 6.5 | 400 | 40 | 150 | 4.22 | 7.40 | 3.90 |

The *HWPCI* is always greater than one because the $L_l$ or $L_r$ value is always greater than the $h$ due to wall irregularities and wavinesses. A *HWPCI* value close to 1 indicates a smooth hole wall profile close to a perfect cylinder geometry.

The Hole Geometry Index *HGI* is an index that combines the values of *HACI* and *HWPCI*, calculated as

$$HGI = HACI \times HWPCI \qquad (3)$$

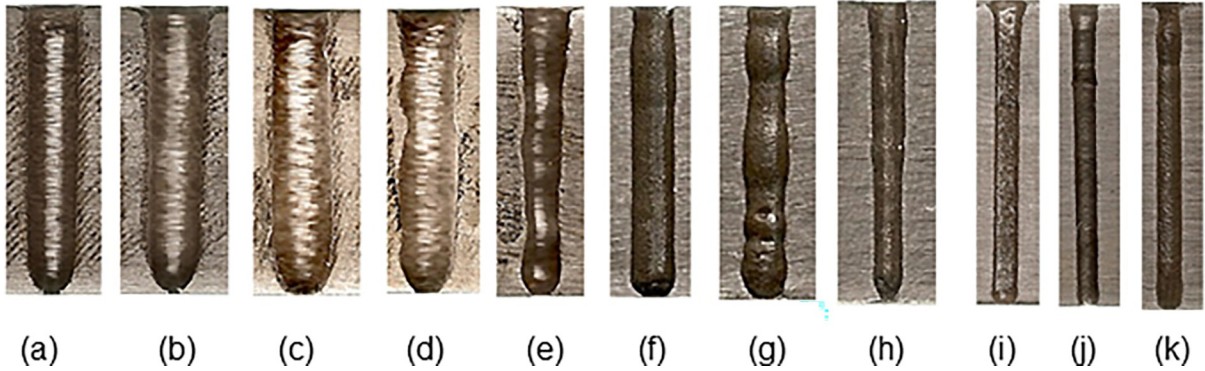

**Fig 3. A hole geometry and terminology ($A_{hole}$: hole cross-sectional area, $d_e$: electrode diameter, $h$: hole depth, $L_r$ and $L_l$: length of the right- and left-side wall profiles of the hole cross-section).**

The *HGI* close to 1 indicates a superior hole geometry, meaning that the hole diameter is close to $d_e$ and the deviation from a straight cylindrical shape is minimal.

The Hole Performance Index *HPI* combines the *HGI* and the ratio of drilling time $t_m$ per unit hole depth $h$ (i.e., the $t_m / h$).

$$HPI = HGI \times \frac{t_m}{h} \qquad (4)$$

If all the measurable parameters are written in an explicit form, then the *HPI* could be given as

$$HPI = \frac{A_{hole}}{A_e} \times \frac{L_l + L_r}{2h} \times \frac{t_m}{h} \qquad (5)$$

A low *HPI* indicates good hole geometry (a low *HGI*) and short drilling time (a low $t_m / h$ ratio).

Example calculations for the indices are given here for the hole section shown in Fig 4G.

$$A_e = d_e \times h = 0.5 \times 8 = 4 \ mm^2$$

$$A_{hole} = 7.39 \ mm^2$$

$$HACI = \frac{A_{hole}}{A_e} = \frac{7.39}{4} = 1.848$$

$$HWPCI = \frac{L_l + L_r}{2h} = \frac{8.29 + 8.33}{2 \times 8} = 1.039$$

$$HGI = HACI \times HWPCI = 1.848 \times 1.039 = 1.920$$

$$HPI = HGI \times \frac{t_m}{h} = 1.920 \times \frac{26}{8} = 6.24$$

The *HPI* formulation includes the two independent indices used by Ayhan et al. [29], HPLC and HCAC, to assess the quality of the hole geometry. In contrast to Ayhan et al.'s indices, the *HPI* is more thorough as it also considers the hole drilling time. The *HPI*,

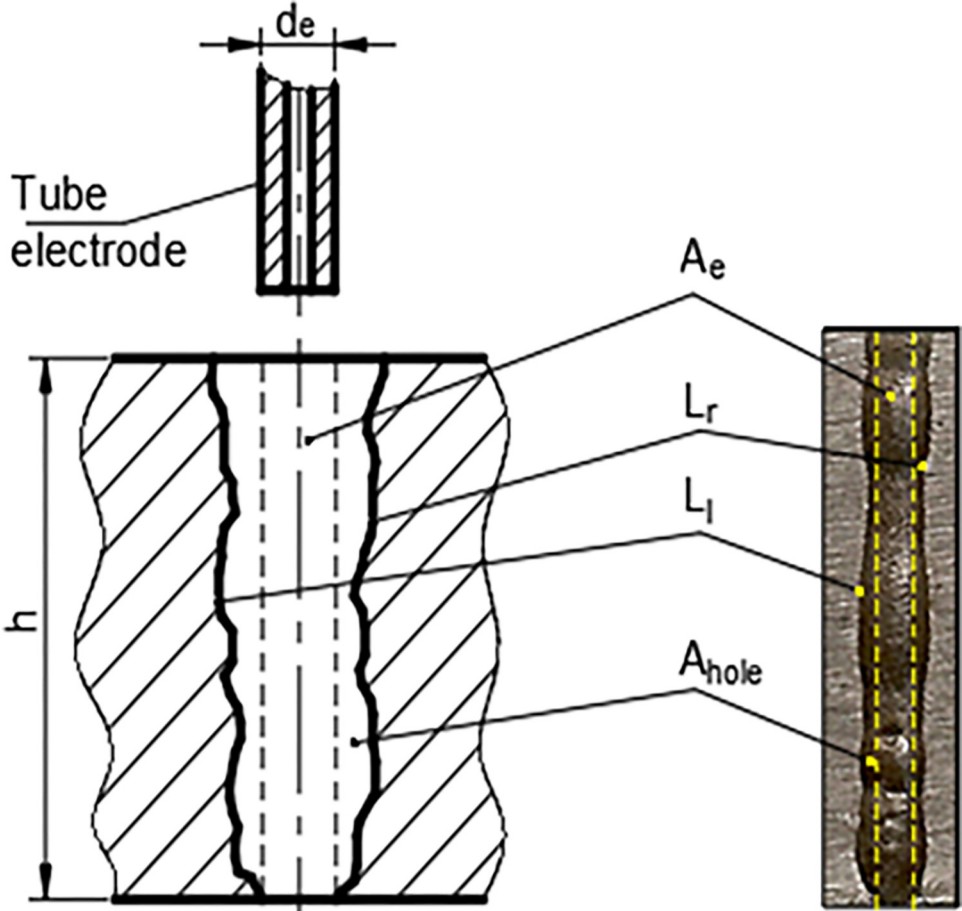

**Fig 4. Sample hole geometries for various machining settings and workpiece materials. a)** Hadfield/4.5 V/200 rpm/ 40 bar/100 g/L, h = 8 mm, **b)** Hadfield/4.5 V/200 rpm/20 bar/125 g/L, h = 8 mm, **c)** Hadfield/4.5 V/200 rpm/20 bar/ 150 g/L, h = 8 mm, **d)** Hadfield/4.5 V/400 rpm/20 bar/150 g/L, h = 8 mm, **e)** Hadfield/5.5 V/400 rpm/20 bar/125 g/L, h = 8 mm, **f)** AISI1040/4.5 V/400 rpm/40 bar/100 g/L, h = 8 mm, **g)** AISI1040/4.5 V/400 rpm/20 bar/150 g/L, h = 8 mm, **h)** AISI1040/6.5 V/200 rpm/20 bar/125 g/L, h = 8 mm, **i)** PM/4.5 V/400 rpm/20 bar/100 g/L, h = 14 mm, **j)** PM/ 5.5 V/400 rpm/40 bar/100 g/L, h = 14 mm, k) PM/6.5 V/200 rpm/40 bar/125 g/L, h = 14 mm.

incorporating the *HACI* value and $t_m/h$ term, presents a more comprehensive evaluation of drilling time compared to the drilling time/tool feed ratio utilized in the HQF formulation by Bilgi et al. [16, 17].

The *HPI* represents the perfection of the hole geometry as well as the drilling time of the holes for different machining settings and workpiece materials. For this reason, the *HPI* was used as the primary and only performance index in this study, as it incorporates all the other performance indices given in Eqs 1–3. The *HPI* values are given in Table 2 for the settings used in the experiments.

**Measurement techniques.** In this study, the $A_e$, $L_r$ and $L_l$ (Fig 3) were calculated using Autocad 2023 software for each hole drilled in the experiment. Firstly, the holes' cross-sectional images were scanned using a high-resolution scanner, and then each image was transferred to the Autocad program in JPEG format. The image was scaled using a known length on the actual sample, after which the right and left-side wall profiles were manually traced to produce splines. Subsequently, the splines were converted to polylines to determine their lengths. The total length of the polylines was then automatically calculated as wall profile

length. For the calculation of $A_e$, the hole entry and exit diameters were drawn with lines and then joined with the $L_r$ and $L_l$ polylines. Subsequently, the area bounded by the $L_r$, $L_l$, hole exit and entry polylines was selected with the "hatch" command, and the $A_e$ value was read from the "properties" tab. All the calculations were transferred to the Excel table with the "list command."

The MIP control unit and synchronized software regulated the electrode feed rate based on the electrolyte's conductivity in the gap and current fluctuations. Parameters such as drilling time, electrode position (drilling depth), machining voltage, and current variations were monitored in real-time and saved individually for each drilling operation through a software interface to an Excel file.

## Results

### Experimental hole geometries

The hole cross-sectional geometries (Fig 4) and drilling times were experimentally obtained for the machining settings and workpiece materials. The findings are as follows:

1. The hole diameter was maximum at the entry section and minimum at the exit section of the workpiece material.

2. Localized diameter variations and non-uniform (typically wavy) wall geometries were observed in some hole sections. These formations were attributed to the agglomeration of the residues from the dissolution reaction and the clogging of the machining zone during the process. This resulted in a short-term increase in the current value and the workpiece removal rate [35]. Another contributing factor to the changes in diameter was the noticeable deterioration of the electrode insulation (coating). No electrode wear was observed in the experiments.

3. In all experiments, the hole diameters were larger than the electrode diameter.

4. Drilling time decreased as the $V_m$, $C_e$, $P_f$ and $n_e$ settings increased for all the three workpiece materials tested.

5. Hole diameters increased for Hadfield and AISI 1040 steels but decreased for PM steel as $n_e$ increased.

6. The hole drilling time of the PM steel was less than that of the AISI 1040 and Hadfield steel samples due to less clogging of the machining zone and fewer interruptions during machining. The Hadfield steel had the longest drilling times.

**Variation of HPI with process parameters.** The study utilized analysis of variance (ANOVA) to determine the statistically significant machining parameters and their respective percent contribution to the *HPI*. Using a loss function, the Taguchi approach [36, 37] computed the divergence between the desired and experimental values. The signal-to-noise ratio, or *S/N* ratio, is then created using this loss function.

There are various *S/N* ratios useable attached to the kind of performance characteristic: lower is better (LB), nominal is best (NB), or higher is better (HB). In ECD, a lower *HPI* indicates better performance. Therefore, LB was selected for the *HPI* to obtain optimum electrochemical drilling performance characteristics. The definitions of the loss function *L* for electrochemical drilling performance outcomes $y_i$ of $n$ repeated numbers for LB are as follows:

$$L_{LB} = \frac{1}{n} \sum_{i=1}^{n} y_i^2 \tag{6}$$

**Table 3. S/N ratios for HPI.**

| Level | M | $V_m$ | $n_e$ | $P_f$ | $C_e$ |
|---|---|---|---|---|---|
| 1 | -15.68 | -19.98 | -17.33 | -17.28 | -18.06 |
| 2 | -21.63 | -16.11 | -16.78 | -16.82 | -16.90 |
| 3 | -13.85 | -15.06 | | | -16.20 |
| Delta | 7.78 | 4.92 | 0.55 | 0.46 | 1.86 |
| Rank | 1 | 2 | 4 | 5 | 3 |

The $S/N_{ij}$ for the $i$'th performance characteristic in the $j$'th experiment can be expressed as;

$$S/N_{ij} = -10.\log(L_{ij}) \tag{7}$$

Irrespective of the performance characteristics category, a higher $S/N$ value indicates better performance. Thus, the highest $S/N$ value represents the optimal process parameter level. The $S/N$ values for each experiment, computed using Eqs 6 and 7, are presented in Table 3 and Fig 5. As illustrated in Fig 5, the optimal performance for the $HPI$ (i.e., the minimum $HPI$) was achieved at level 3 for the material $M$ (PM steel), at level 3 for the $V_m$ (6.5 V), at level 2 for the $n_e$ (400 rpm), at level 2 for the $P_f$ (40 bar), and at level 3 for the $C_e$ (150 g/L).

Table 3 also shows the difference (Delta) between the process parameters' maximum and minimum $S/N$ ratios: 7.78 for $M$, 4.92 for $V_m$, 1.86 for $C_e$, 0.55 for $n_e$, and 0.46 for $P_f$. The comparison of these values indicates that the highly effective controllable factors on $HPI$ were the

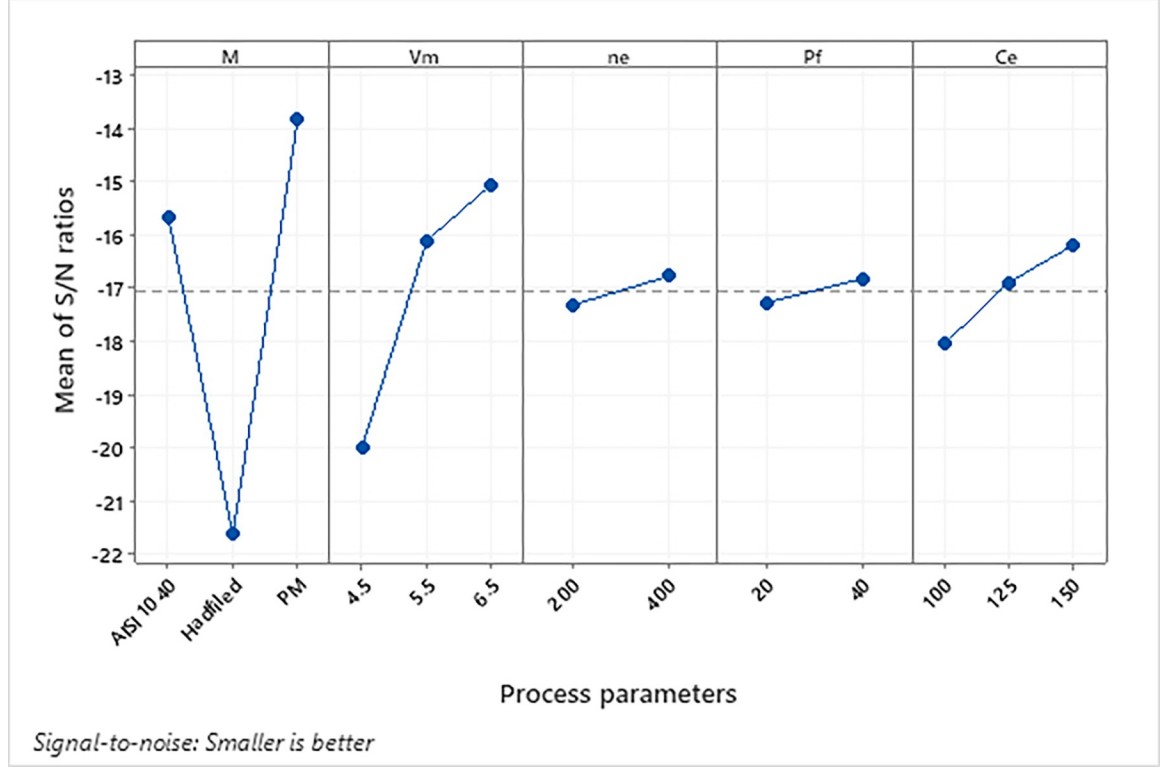

**Fig 5. The effect of process parameters on HPI.**

**Table 4. Result of ANOVA for *HPI*.**

| Source | DF | Seq SS | Contribution | Adj SS | Adj MS | *F*-value | *p*-value |
|---|---|---|---|---|---|---|---|
| $M$ | 2 | 1322.28 | 51.92% | 1322.28 | 661.139 | 116.59# | 0.000 |
| $V_m$ | 2 | 594.56 | 23.34% | 594.56 | 297.278 | 52.42# | 0.000 |
| $n_e$ | 1 | 2.31 | 0.09% | 2.31 | 2.307 | 0.41 | 0.525 |
| $P_f$ | 1 | 10.75 | 0.42% | 10.75 | 10.748 | 1.90 | 0.172 |
| $C_e$ | 2 | 55.67 | 2.19% | 55.67 | 27.834 | 4.91# | 0.009 |
| Error | 99 | 561.40 | 22.04% | 561.40 | 5.671 | | |
| Total | 107 | 2546.95 | 100.00% | | | | |

# at least 99% confidence

$M$, with a maximum value of 7.78 (Rank 1) and $V_m$, with a value of 4.92 (Rank 2). In order of importance, the other factors affecting the *HPI* are $C_e$, $n_e$ and $P_f$.

The *F*-test allows one to decide with a certain level of confidence whether the estimates are significantly different [36, 37]. A bigger *F* value designates that the change in process parameters remarkably affects performance characteristics. Therefore, the greater the *F*-statistic, the more substantial the evidence of a variation the group means. The relevant confidence table is compared with the *F* values of the process parameters. When the $F_{\alpha,v1,v2}$ value in the confidence table are less than the *F* value of the process parameter, this statistically means that the process parameter affects performance characteristics at a very high confidence level. Here, $\alpha$ is risk, $v_1$ and $v_2$ are degrees of freedom regarding the numerator and denominator.

Based on the ANOVA results (Table 4), the statistical significance of the difference between group means can be determined by examining the corresponding *p*-value for the *F*-statistic. If the *p*-value is less than 0.05, the null hypothesis of ANOVA is rejected and it is concluded that there is a statistically significant difference between the means of the groups. If the *p*-value is bigger than or equal to 0.05, the null hypothesis is not rejected and it is concluded that there is insufficient evidence to demonstrate a statistically significant difference between the means of the groups. In this study, the *p*-value is 0.525 for electrode rotational speed and 0.172 for electrolyte flushing pressure. This indicates insufficient evidence to conclude that the group means for electrode rotational speed and electrolyte concentration differ statistically significantly.

Table 4 also shows the percentage contributions of the electrochemical drilling parameters to the *HPI*. The percent contribution is obtained by summing all the sum of squares (SS) and then taking each SS, dividing by the total SS and multiplying by 100. The primary factor affecting the *HPI* was the workpiece material, with a 51.92% contribution, followed by the machining voltage, with a 23.34% contribution. The electrolyte concentration, injection pressure, and electrode rotational speed contributions were much lower (2.19%, 0.42%, and 0.09%, respectively). The high contribution of the workpiece material and machining voltage on *HIP* are consistent with Faraday's material removal rate (*MRR*) law [38] for electrochemical machining,

$$MRR = C \times A \times \frac{I}{Z \times d} \tag{8}$$

where $C$ is a constant, $I$ is the machining voltage, and $A$, $Z$ and $d$ are the workpiece material's atomic weight, valence and density. From the equation, it is clear that the drilling time (i.e., $t_m/h$ term in the *HPI*) is highly dependent on the workpiece material type and the machining voltage.

## Evaluation of experimental findings and HPI

According to Eq 5, the *HPI* is directly proportional to $A_{hole}$, $L_r + L_l$, and $t_m$, while $A_e$ and $h$ values remain constant throughout the experiments. Therefore, to minimize the *HPI*, $A_{hole}$, $L_r$, $L_l$, and $t_m$ values should be low. Additionally, the *S/N* ratio analysis (Fig 5) revealed that the optimum process parameter combination for the minimum *HPI* was the highest settings of the machining voltage, electrode rotational speed, electrolyte injection pressure and electrolyte concentration, and the use of PM steel workpiece. Thus, it is recommended to use the maximum settings for the machining parameters and the PM steel workpiece to minimize $A_{hole}$, $L_r$, $L_l$, and $t_m$ values. Smaller $A_{hole}$, $L_r$, and $L_l$ values represent better hole geometries, while lower $t_m$ values indicate shorter drilling times.

It has been shown that the lowest $t_m$ was achieved when the machining parameters were set at their highest level for the three workpiece materials, as previously mentioned in experimental finding #4 in Section "Experimental hole geometries". Moreover, the *HGI*, which directly influences the *HPI* value as depicted in Eq 3, plays a crucial role in evaluating the correspondence between the hole diameter and electrode diameter ($d_e$) and the level of deviation from a perfectly cylindrical shape. Images of the holes, with a depth of 14 mm, that were drilled in the PM part are displayed in Fig 4İ–4K. The hole shown in Fig 4K has the lowest *HPI* value (3.94), while the hole in Fig 4J ranks second with an *HPI* of 4.07 (Table 2). Given the similarities in the geometries of the two holes, it can be inferred that the drilling times are also close. This is supported by the observed drilling times of 28 minutes for the hole in Fig 4J and 24 minutes for the hole in Fig 4K. The hole in Fig 4İ has an *HGI* value (2.37) that resembles those in Fig 4J and 4K. However, its higher *HPI* value (9.83) indicates a longer drilling time (58 minutes) than the other two. Fig 4A–4D depict the holes with a depth of 8 mm that were drilled in Hadfield steel. These holes' close *HGI* values (2.78, 3.38, 3.73, and 3.62) confirm their geometric similarities. These holes' high *HPI* values (20.59, 22.82, 21.04, and 20.41, respectively) suggest that they required long drilling times (59 min, 54 min, 45 min, and 45 min, respectively). On the other hand, the hole drilled in the same material (Fig 4H) with a much lower *HGI* (1.72) and *HPI* (8.61) implies a superior hole geometry and shorter drilling time (32 minutes) compared to the others.

So, the experimental findings are represented in the *HPI* in terms of $A_{hole}$, $L_r$, $L_l$, and $t_m$. For example,

- "localized diameter variations and non-uniform wavy wall geometries" are represented in terms of $A_{hole}$, $L_r$ and $L_l$,

- "hole diameters larger than the electrode diameter" and "hole diameters increased for Hadfield and AISI 1040 steels but decreased for PM steel with increasing electrode rotational speed" are represented in terms of $A_{hole}$,

- "the drilling time decreased as the machining voltage, electrolyte concentration, electrode flushing pressure, and electrode rotational speed increased", "the hole drilling time of the

**Table 5. The validation experiment for *HPI*.**

| | Starting machining parameters | Optimal machining parameters | |
| --- | --- | --- | --- |
| | | **Prediction** | **Experiment** |
| Level | $M2V_m2n_e2P_f2C_e2$ | $M3V_m3n_e2P_f2C_e3$ | $M3V_m3n_e2P_f2C_e3$ |
| *HPI* | 9.9360 | 1.9037 | 3.9003 |
| *S/N* ratio for *HPI* (dB) | -19.9442 | -5.5919 | -11.8219 |

Improvement *S/N* ratio for *HPI* = 8.12 dB

PM steel was less than that of the AISI 1040 and Hadfield steel samples," and "the Hadfield steel had the longest drilling times" are represented in terms of $t_m$ in the *HPI*.

These findings demonstrate that the *HPI* is a suitable tool for modeling experimental results related to hole geometry and drilling time.

**Validation experiments.** The validation experiment tests a particular combination of the factors and levels previously assessed to validate the conclusions drawn during the analysis phase. This paper reports on a novel experiment planned and carried out to identify the ideal levels of machining parameters. The last phase involves predicting and verifying the enhancement of performance characteristics. The predicted *S/N* ratio $\hat{\eta}$ can be calculated using the ideal levels of electrochemical drilling parameters.

$$\hat{\eta} = \eta_m + (\sum_{i=1}^{k}(\bar{\eta}_i - \eta_m) \tag{9}$$

Here, the $S/N_m$ is the total mean of the *S/N* ratio, $\bar{\eta}_i$ is the mean of the *S/N* ratio at the ideal level, and $k$ is the number of the main process factors significantly affecting performance.

Table 5 presents the results of experimental verification using optimal electrochemical drilling factors. It compares the predicted *HPI* with the actual *HPI* using the optimal process factors. The *S/N* ratio for *HPI* improved by 8.12 dB from the starting electrochemical drilling factors to the optimal ones, while the *HPI* decreased by 2.54 times. The approach used significantly improved the *HPI*. The experimental results have validated the currency of the Taguchi approach for improving electrochemical drilling performance and optimizing electrochemical drilling factors.

## Conclusion

This study presents a new Hole Performance Index (HPI) that incorporates electrode diameter, hole depth, hole cross-sectional area, wall profile lengths on both sides, and drilling time to evaluate the quality of hole geometry and drilling efficiency in the ECD process. The HPI demonstrated its utility as a metric for assessing these factors across various machining parameters and workpiece materials. Lower HPI values indicate enhanced hole wall profiles, improved alignment with the electrode area, and reduced drilling durations.

Signal-to-noise ratio analysis indicated that the optimal process parameters for minimizing HPI included elevated settings for machining voltage, electrode rotational speed, electrolyte injection pressure, electrolyte concentration, and using a PM steel workpiece. Furthermore, the workpiece material and machining voltage significantly impacted HPI, contributing 51.92% and 23.34%, respectively. The contributions of the electrode rotational speed, electrolyte injection pressure, and electrolyte concentration to HPI were 2.19%, 0.42%, and 0.09%, respectively, indicating their limited effectiveness.

The proposed statistical technique decreased HPI by 2.2, transitioning from initial machining parameters to optimal settings. This illustrates the efficacy of the Taguchi method in improving machining efficiency and optimizing ECD operations.

Future research may investigate reduced settings for electrode rotational speed, electrolyte injection pressure, and electrolyte concentration to optimize hole geometry and drilling speed while minimizing energy and salt consumption, given that these parameters demonstrated limited influence on HPI in this study.

## Author Contributions

**Conceptualization:** C. Çoğun.

**Data curation:** C. Çoğun.

**Formal analysis:** C. Çoğun, Robert Cep.

**Funding acquisition:** Robert Cep.

**Investigation:** N. Tosun, H. B. Özerkan, C. Çoğun.

**Methodology:** N. Tosun, H. B. Özerkan, C. Çoğun.

**Project administration:** C. Çoğun.

**Resources:** H. B. Özerkan, C. Çoğun.

**Software:** N. Tosun, Robert Cep.

**Supervision:** C. Çoğun.

**Validation:** Robert Cep.

**Visualization:** C. Çoğun.

**Writing – original draft:** N. Tosun, H. B. Özerkan, C. Çoğun, Robert Cep.

**Writing – review & editing:** N. Tosun, C. Çoğun.

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
