## [Decision Letter · Decision Letter 0]

18 Jul 2024

PONE-D-24-24552A NOVEL HOLE PERFORMANCE INDEX TO EVALUATE THE HOLE GEOMETRY AND DRILLING TIMEIN THE ELECTROCHEMICAL DRILLING PROCESSPLOS ONE

Dear Dr. Özerkan,

Thank you for submitting your manuscript to PLOS ONE. After careful consideration, we feel that it has merit but does not fully meet PLOS ONE’s publication criteria as it currently stands. Therefore, we invite you to submit a revised version of the manuscript that addresses the points raised during the review process.

ACADEMIC EDITOR: I have now received sufficient Reviewers' feedback regarding your submitted paper. The paper need to go through major changes as per their views. If Authors willing to take the changes mentioned and make a point to point reply then I will be happy to reconsider my decision. Authors must take all the comments seriously and upgrade their paper.

We look forward to receiving your revised manuscript.

Kind regards,

Himadri Majumder, Ph.D

Academic Editor

PLOS ONE

Journal Requirements:

3. Thank you for stating the following financial disclosure: "The author, R.V., would like to thank the National Research Foundation of Korea (NRF), which was funded by the Ministry of Education".

4. We note that your Data Availability Statement is currently as follows: "All relevant data are within the manuscript and its Supporting Information files."

Additional Editor Comments:

I have now received sufficient Reviewers' feedback regarding your submitted paper. The paper need to go through major changes as per their views. If Authors willing to take the changes mentioned and make a point to point reply then I will be happy to reconsider my decision. Authors must take all the comments seriously and upgrade their paper.

Reviewers' comments:

Reviewer's Responses to Questions

**Comments to the Author**

1. Is the manuscript technically sound, and do the data support the conclusions?

Reviewer #1: No

Reviewer #2: Yes

Reviewer #3: Partly

Reviewer #4: Yes

2. Has the statistical analysis been performed appropriately and rigorously? 

Reviewer #1: Yes

Reviewer #2: Yes

Reviewer #3: No

Reviewer #4: Yes

3. Have the authors made all data underlying the findings in their manuscript fully available?

Reviewer #1: Yes

Reviewer #2: Yes

Reviewer #3: No

Reviewer #4: Yes

4. Is the manuscript presented in an intelligible fashion and written in standard English?

Reviewer #1: Yes

Reviewer #2: Yes

Reviewer #3: No

Reviewer #4: Yes

5. Review Comments to the Author

Reviewer #1: Authors are advised to see the comments given in the attachement and revise the manuscript accordingly. Actually it is not written properly and proper revision is advised otherwise it will not be accepted in the publication.

Reviewer #2: The authors have done a good work and its focus on dimension analyses of the machined hole. There are few queries authors need to clarify

Replace the figures with sharp resolution, figure 1 is blurred.

Do you have SEM image of drilled hole, add the same.

what is the thickness of the workpiece is all material of same size?

what is purpose of measuring the time and i don't see any output performance value related to time?

How did you ensure the completion of hole drilling process completion?

How did you measure the hole circumference waviness?

Provide the details of instruments for evaluation of output performance?

Based on ANOVA you suggest material and voltage has significant influence. Why? Based on the ECM principle the material dissolution takes place at faster rate irrespective of the material. Justify with literature.

Add the reference with

doi.org/10.3103/S1068375517050143

doi.org/10.1080/10426914.2022.2030874,

10.4314/bcse.v37i5.17

Reviewer #3: 1. What material is the electrode made of (chemical composition)?

2. What shape does the electrode have (dimensions)?

3. How did the authors analyze the thermal changes and cracks that occurred?

4. Please show a diagram of electrochemical hole machining (ESD).

5. What electrolytes were used in the studies?

6. How to measure Lr and Ll? Please insert a drawing. How was the hole axis determined?

7. Please describe the methodology for measuring the hole.

8. How did the authors take into account the influence of electrode wear?

9. How many repetitions of the study were performed by the authors? What research plan did the authors adopt?

10. Please compare the proposed indicator with indicators described in the literature. It is necessary.

11. How will the adopted HPI distinguish the slenderness of the hole from the ovality?

Reviewer #4: The author has done the research entitled “A NOVEL HOLE PERFORMANCE INDEX TO EVALUATE THE HOLE GEOMETRY AND DRILLING TIME IN THE ELECTROCHEMICAL DRILLING PROCESS”. This paper examines the impact of process variables, such as machining voltage, electrolyte concentration, electrode rotational speed, electrolyte flushing pressure, and workpiece material, on the novel hole performance index HPI in the electrochemical drilling process. The manuscript has strong technical content. The tests were also carried out scientifically. With major changes, this manuscript will be accepted. The following major corrections must be made.

1. The expansion of HGI is not listed in the nomenclature section which may confuse the reader. It should be added.

2. In Table 2, some of the process parameters are taken into account as two levels, some are three levels. Justify in depth the selection of process parameters and their corresponding levels, providing a comprehensive explanation.

3. In Table 3, the remarks column does not explore valuable data. It should be thoroughly revised and rewritten with some statements. If it is not the most important, you can delete that particular column.

4. In the result and discussion, the author has discussed only HPI, not HGI. Justify.

5. All figures are not clear and the quality of the figure is very poor.

6. How to find out the percentage of contributions for each process parameter in Table 5. Also, the process parameters like ne, Pf, and Ce have not contributed much. Justify.

7. The author should include the microstructural images (OM or SEM) at least the optimized conditions.

6. PLOS authors have the option to publish the peer review history of their article (what does this mean?). If published, this will include your full peer review and any attached files.

Reviewer #1: No

Reviewer #2: No

Reviewer #3: **Yes: **Krzysztof Szwajka

Reviewer #4: **Yes: **Siva Kumar Mahalingam

---

## [Author Response · Author response to Decision Letter 0]

25 Aug 2024

Dear Reviewers,

We have made all necessary changes and improvements to the paper per the yours suggestions and critique. We thank all reviewers for their efforts and suggestions. We have modified the paper, and the detailed corrections. Responses to Reviewer 1's evaluation are written on the reviewer PDF file. The responses to Reviewers 2, 3, and 4 are written in a table opposite each evaluation in a doc.x file.

The manuscript has been resubmitted to the journal. 

Thank you for your all contributions.

Best Regards,

---

## [Decision Letter · Decision Letter 1]

11 Sep 2024

PONE-D-24-24552R1A NOVEL HOLE PERFORMANCE INDEX TO EVALUATE THE HOLE GEOMETRY AND DRILLING TIMEIN THE ELECTROCHEMICAL DRILLING PROCESSPLOS ONE

Dear Dr. Özerkan,

Thank you for submitting your manuscript to PLOS ONE. After careful consideration, we feel that it has merit but does not fully meet PLOS ONE’s publication criteria as it currently stands. Therefore, we invite you to submit a revised version of the manuscript that addresses the points raised during the review process.

**ACADEMIC EDITOR: ** As per the reviewer feedback, the authors didn't care to comply the suggestion provided by the reviewer. So I am recommending "Major revision". Authors must reply all the concerned raised and reply accordingly.

We look forward to receiving your revised manuscript.

Kind regards,

Himadri Majumder, Ph.D

Academic Editor

PLOS ONE

Additional Editor Comments:

As per the reviewer feedback, the authors didn't care to comply the suggestion provided by the reviewer. So I am recommending "Major revision". Authors must reply all the concerned raised and reply accordingly.

Reviewers' comments:

Reviewer's Responses to Questions

**Comments to the Author**

1. If the authors have adequately addressed your comments raised in a previous round of review and you feel that this manuscript is now acceptable for publication, you may indicate that here to bypass the “Comments to the Author” section, enter your conflict of interest statement in the “Confidential to Editor” section, and submit your "Accept" recommendation.

Reviewer #1: (No Response)

Reviewer #2: All comments have been addressed

Reviewer #3: All comments have been addressed

2. Is the manuscript technically sound, and do the data support the conclusions?

Reviewer #1: No

Reviewer #2: Yes

Reviewer #3: Yes

3. Has the statistical analysis been performed appropriately and rigorously? 

Reviewer #1: I Don't Know

Reviewer #2: I Don't Know

Reviewer #3: Yes

4. Have the authors made all data underlying the findings in their manuscript fully available?

Reviewer #1: No

Reviewer #2: Yes

Reviewer #3: Yes

5. Is the manuscript presented in an intelligible fashion and written in standard English?

Reviewer #1: No

Reviewer #2: Yes

Reviewer #3: Yes

6. Review Comments to the Author

Reviewer #1: i have suggested major comments to the author but i dint find any comments with revision in the paper. So i am advising authors to read the comments and revise paper accordingly.

Reviewer #2: The authors have addressed the all comments.The queries are answered and can be accepted for possible publication.

It can be ACCEPTED

Reviewer #3: (No Response)

7. PLOS authors have the option to publish the peer review history of their article (what does this mean?). If published, this will include your full peer review and any attached files.

Reviewer #1: No

Reviewer #2: No

Reviewer #3: **Yes: **Krzysztof Szwajka

---

## [Author Response · Author response to Decision Letter 1]

14 Oct 2024

Reviewer #1: Comments Author responses

1- (For abstract) 

very long sentences...authors are advised to see the entire mansucript and revise it properly...such long sentences must not be in the paper Thank you. The long sentences are revised.

2-

 (For nomenclature) 

see nomenclature section properly.. Thank you. We reworked the "nomenclature" section.

3- (For introduction part) 

No reference...also objectives must be written properly and it must not be in first paragraph.. The simple/basic information given in the first paragraph is a well-known fact about the ECD process for the researchers and readers in the field. So, with all our respect, the authors prefer not to give reference(s) in this part.

A separate section, named "objectives," is added to the manuscript following the "introduction" section.

4-

 (For literature survey findings)

it is not written properly... Thank you. This part is rewritten.

5- (For experiments part)

is it proper...where is materials and method section... Thank you for your constructive criticism. The "materials and methods" section is added.

6- (For Fig 3.)

 needs to be revised...explanation is not proper The caption is rewritten.

7- 

(For “Confirmation experiments part”)

wrong one...in place of confirmation, valdiation is a proper word and need to be explained properly about the variations too..authors are instructed to revise properly... The "confirmation" wording is an almost standart usage for the ANOVA and related statistical analyses. Anyway, we have changed the word "confirmation" to "validation".

8- 

(For “Conclusion” part)

Conclusion is poor...authors are instructed to add the conclusion properly..it looks like pargraph The conclusion section is reworked.

---

## [Decision Letter · Decision Letter 2]

23 Oct 2024

PONE-D-24-24552R2A NOVEL HOLE PERFORMANCE INDEX TO EVALUATE THE HOLE GEOMETRY AND DRILLING TIMEIN THE ELECTROCHEMICAL DRILLING PROCESSPLOS ONE

Dear Dr. Özerkan,

Thank you for submitting your manuscript to PLOS ONE. After careful consideration, we feel that it has merit but does not fully meet PLOS ONE’s publication criteria as it currently stands. Therefore, we invite you to submit a revised version of the manuscript that addresses the points raised during the review process.

We look forward to receiving your revised manuscript.

Kind regards,

Himadri Majumder, Ph.D

Academic Editor

PLOS ONE

Journal Requirements:

Reviewers' comments:

Reviewer's Responses to Questions

**Comments to the Author**

1. If the authors have adequately addressed your comments raised in a previous round of review and you feel that this manuscript is now acceptable for publication, you may indicate that here to bypass the “Comments to the Author” section, enter your conflict of interest statement in the “Confidential to Editor” section, and submit your "Accept" recommendation.

Reviewer #1: (No Response)

2. Is the manuscript technically sound, and do the data support the conclusions?

Reviewer #1: Partly

3. Has the statistical analysis been performed appropriately and rigorously? 

Reviewer #1: No

4. Have the authors made all data underlying the findings in their manuscript fully available?

Reviewer #1: No

5. Is the manuscript presented in an intelligible fashion and written in standard English?

Reviewer #1: Yes

6. Review Comments to the Author

Reviewer #1: see the comments given during first revision and revise properly. Still abstract and conclusion have not rewritten properly.

7. PLOS authors have the option to publish the peer review history of their article (what does this mean?). If published, this will include your full peer review and any attached files.

Reviewer #1: No

---

## [Author Response · Author response to Decision Letter 2]

1 Nov 2024

Reviewer 1 Comments: (For abstract and conclusion) 

see the comments given during first revision and revise properly. Still abstract and conclusion have not rewritten properly.

Response:

Thank you. The comments in the first revision have been revised properly. The summary and conclusion sections have been revised and rewritten.

---

## [Decision Letter · Decision Letter 3]

7 Nov 2024

A NOVEL HOLE PERFORMANCE INDEX TO EVALUATE THE HOLE GEOMETRY AND DRILLING TIMEIN THE ELECTROCHEMICAL DRILLING PROCESS

PONE-D-24-24552R3

Dear Dr. Özerkan,

We’re pleased to inform you that your manuscript has been judged scientifically suitable for publication and will be formally accepted for publication once it meets all outstanding technical requirements.

Kind regards,

Himadri Majumder, Ph.D

Academic Editor

PLOS ONE

Additional Editor Comments (optional):

As per the reviewer feedback, I am recommending "Acceptance" for this revised paper.

Reviewers' comments:

Reviewer's Responses to Questions

**Comments to the Author**

1. If the authors have adequately addressed your comments raised in a previous round of review and you feel that this manuscript is now acceptable for publication, you may indicate that here to bypass the “Comments to the Author” section, enter your conflict of interest statement in the “Confidential to Editor” section, and submit your "Accept" recommendation.

Reviewer #1: All comments have been addressed

2. Is the manuscript technically sound, and do the data support the conclusions?

Reviewer #1: Yes

3. Has the statistical analysis been performed appropriately and rigorously? 

Reviewer #1: Yes

4. Have the authors made all data underlying the findings in their manuscript fully available?

Reviewer #1: Yes

5. Is the manuscript presented in an intelligible fashion and written in standard English?

Reviewer #1: Yes

6. Review Comments to the Author

Reviewer #1: All the comments have been successfully incorporated in the paper. It is accepted for the publication in the Plos one.

7. PLOS authors have the option to publish the peer review history of their article (what does this mean?). If published, this will include your full peer review and any attached files.

Reviewer #1: No

---

## [Editor Report · Acceptance letter]

20 Nov 2024

PONE-D-24-24552R3 

PLOS ONE

Dear Dr. Özerkan, 

I'm pleased to inform you that your manuscript has been deemed suitable for publication in PLOS ONE. Congratulations! Your manuscript is now being handed over to our production team.

Kind regards, 

on behalf of

Dr. Himadri Majumder 

Academic Editor

PLOS ONE